# Engineering electronic structure to prolong relaxation times in molecular qubits by minimising orbital angular momentum

Ana-Maria Ariciu[1,2,4], David H. Woen[3,4], Daniel N. Huh [3], Lydia E. Nodaraki[1], Andreas K. Kostopoulos[1], Conrad A.P. Goodwin [1], Nicholas F. Chilton [1], Eric J.L. McInnes [1,2], Richard E.P. Winpenny [1], William J. Evans[3] & Floriana Tuna[1,2]

The proposal that paramagnetic transition metal complexes could be used as qubits for quantum information processing (QIP) requires that the molecules retain the spin information for a sufficient length of time to allow computation and error correction. Therefore, understanding how the electron spin-lattice relaxation time ($T_1$) and phase memory time ($T_m$) relate to structure is important. Previous studies have focused on the ligand shell surrounding the paramagnetic centre, seeking to increase rigidity or remove elements with nuclear spins or both. Here we have studied a family of early 3d or 4f metals in the +2 oxidation states where the ground state is effectively a $^2S$ state. This leads to a highly isotropic spin and hence makes the putative qubit insensitive to its environment. We have studied how this influences $T_1$ and $T_m$ and show unusually long relaxation times given that the ligand shell is rich in nuclear spins and non-rigid.

[1] School of Chemistry, The University of Manchester, Oxford Road, Manchester M13 9PL, UK. [2] Photon Science Institute, The University of Manchester, Oxford Road, Manchester M13 9PL, UK. [3] Department of Chemistry, University of California, Irvine, CA 92697-2025, USA. [4]These authors contributed equally: Ana-Maria Ariciu, David H. Woen. Correspondence and requests for materials should be addressed to R.E.P.W. (email: richard. winpenny@manchester.ac.uk) or to W.J.E. (email: wevans@uci.edu) or to F.T. (email: floriana.tuna@manchester.ac.uk)

L euenberger and Loss first proposed using electron spins within molecules for quantum information processing (QIP)[1] by showing how the Grover algorithm could be mapped onto a $Mn_{12}$ cage[2]. Since then several groups have examined two-level ($S = 1/2$) molecular spin systems as possible qubits[3–12] and the Grover algorithm has been implemented in a four-level molecular qudit[13]. Molecular electron spin qubits have an obvious disadvantage: as they are molecules, the magnetic moments of nuclear spins and molecular vibrations will produce noise that could cause electron spin relaxation and therefore loss of phase information within the qubit before computations can be performed. Studies to address this disadvantage include use of advanced microwave pulse sequences[3] to repair the damage done by noise and hence increase phase memory times ($T_m$). A second approach uses spin transitions that do not vary with magnetic field at first order[12], often called clock transitions; the lack of dependence on local field renders the qubit insensitive to its environment and hence lengthens $T_m$.

The magnetic noise in molecules is dominated at low temperatures by nearby nuclear spins undergoing flip-flop processes causing an oscillating magnetic field: reducing this source of noise has been widely studied to engineer better qubits[7–11]. Removal of nuclear spins from the qubit and/or embedding the molecule in a nuclear-spin-free lattice has led to phase memory times ($T_m$) long enough to permit coherent spin manipulation via Rabi oscillations[9–11], sometimes even at room temperature[11]. However, even in the absence of nuclear spins, $T_m$ becomes limited by spin-lattice relaxation ($T_1$) that is strongly temperature dependent. $T_1$ is dominated by modulation of spin-orbit coupling via molecular vibrations. It is the spin-orbit coupling that mixes spin and orbital angular momentum, hence enabling exchange of energy between the spin system and the lattice when the electronic energy levels are modulated. Reducing the impact of molecular vibrations is a challenge. The intuitive approach is to reduce the influence of molecular vibrations by making a more rigid molecule or lattice, which does lead to longer phase memory times by lengthening $T_1$[11].

We noted a study where $Zn^+$ ions, encapsulated in a metal-organic framework, show phase memory times of 2 µs at room temperature[14]. This is ascribed to the unpaired electron being in a $^2S$ state and hence with minimal anisotropy. Therefore, we sought a molecular species that would mimic the highly isotropic nature of these embedded $Zn^+$ ions. All molecular qubits proposed thus far have the unpaired electron spin in a p-, d- or f-orbital and have little s-character. Our target was a molecule with a $^2S$ ground state.

We identified a family of molecules that might exemplify this strategy. The compounds are three-coordinate with local pseudo-$C_3$ symmetry at the metal ion. In this point group, the lowest energy d-orbital is $d_{z^2}$ and it has the same irreducible representation as an s-orbital; therefore, the s- and $d_{z^2}$ orbitals will mix and regardless of the relative admixture of the two atomic orbitals, an S-like state will result as both orbitals have $m_l = 0$. The compounds contain early d-block or 4f-block metals in the +2 oxidation state, which is very unusual for the metals involved. The four compounds are: [K(2.2.2-cryptand)][Y(Cp′)₃] **1** (Cp′ = $C_5H_4SiMe_3$)[15,16] (Supplementary Fig. 1); [K(2.2.2-cryptand)][Lu(Cp′)₃] **2**; [K(2.2.2-cryptand)][La(Cp′)₃] **3**; and [K(2.2.2-cryptand)][Sc{(N(SiMe₃)₂}₃] **4**. These complexes flout the established criteria[17] for long $T_1$ or $T_m$ in molecular qubits, as they are far from rigid and contain numerous hydrogen atoms and multiple methyl groups, whose rotation is a known decoherence mechanism[18]. Yet here we report coherent spin manipulations on **1** at room temperature. We achieve this by engineering the electronic configuration of the metal centre, an approach that could be combined with the beautiful engineering of the ligand shell demonstrated by other groups[7–11].

## Results

**Electron paramagnetic resonance spectroscopy.** In compounds **1–4** the metal centre in the anion is three-coordinate with a pseudo-$C_3$ axis passing through the metal, perpendicular to the plane defined by the Cp′ centroids in **1–3** and the N atoms in **4** (Fig. 1a). In all cases the continuous-wave EPR (CW-EPR) spectroscopy of a fluid solution of these compounds shows a multiplet due to hyperfine coupling ($A$) of the single unpaired electron to the metal nucleus (Supplementary Fig. 2; Table 1). On freezing the solutions, a small axial $g$-anisotropy is observed in all cases; where z is coincident with the pseudo-$C_3$ axis), with a near isotropic hyperfine coupling (Fig. 2a; Table 1). The highly isotropic hyperfine coupling and minimal $g$-anisotropy suggests that the electron is in an S-like state.

We focus here on **1** as it has an isostructural diamagnetic analogue [K(2.2.2-cryptand)][Yb(Cp′)₃] **5**, and therefore it can be doped into **5** to yield a paramagnetically dilute crystalline sample. The studies described below were performed on both frozen solutions of **1** and on single crystals of **1@5** at a doping level of approximately 2%. Probing the spin dynamics of **1** with pulsed EPR methods, echo-detected field-swept EPR (EDFS; see Fig. 2a and Supplementary Fig. 2) spectra[19] of **1** in THF are entirely consistent with CW-EPR; these spectra persist up to temperatures limited only by thawing of the solvent. In order to investigate the spin dynamics of **1** at higher temperatures, we studied a single crystal of **1@5** which gives identical g and $A^Y$ parameters to the frozen THF solution of **1**, and confirms the assignment of the principal directions of $g_z$ and $g_{x,y}$ with respect to the molecular structure (Fig. 2b, Supplementary Figs. 3 and 4). For a single crystal of **1@5**, we find spin echoes between 5 and 300 K.

Relaxation measurements (Supplementary Figs. 5–9) show that $T_1$ is ~10–20 ms below 10 K, but remains as long as 2 µs at 300 K, while $T_m$ varies between 2 and 0.4 µs in the same temperature range (Fig. 2e; Supplementary Tables 1-4). $T_1$ and $T_m$ are very similar between frozen solution and single crystal measurements, showing that the spin dynamics of **1** are independent of the phase of the sample; this is also remarkable because the THF ($C_4H_8O$) solvent is rich with $^1H$ nuclear spins. Furthermore, $T_1$ and $T_m$ are also largely independent of molecular orientation or which nuclear spin $m_I$ transition is monitored (see different observer positions in Fig. 2). $T_1$ has a Raman-like dependence on temperature above 20 K ($T_1 = C^{-1}T^{-n}$; $C = 1.3(3) \times 10^{-8} \mu s^{-1}$ $K^{-n}$, $n = 3.16(5)$), and only a very weak temperature dependence below 20 K. $T_m$ has a much weaker temperature dependence than $T_1$, and, due to the coincidence of the data for frozen solution and single crystal phase, we suspect that $T_m$ is limited by nuclear spin diffusion given the $^1H$-rich environment of the molecule. (There is a minimum in $T_m$ around 80 K due to methyl-group rotations.) Importantly, at room temperature $T_m$ is smaller than $T_1$ by a factor of 4–5 and hence $T_1$ has not limited $T_m$ and spin echoes are still observable.

Given the long $T_m$ times, we are able to perform coherent spin manipulations as demonstrated by observation of Rabi oscillations[20] even at room temperature for **1@5** in transient nutation experiments (Fig. 2c and Supplementary Figs. 11–22). These periodic oscillations in echo intensity as a function of the duration of a preceding nutation pulse correspond to cyclical generation of $|\pm m_S\rangle$ superposition states. All oscillations show the characteristic linear dependence of Rabi frequency ($\Omega_R$) on nutation pulse attenuation (Fig. 2d and Supplementary Figs. 11–23). $\Omega_R$ is independent of temperature and the $m_I$ transition being monitored (Supplementary Figs. 16 and 23), and is very similar between **1@5** and a frozen THF solution of **1** (Supplementary Figs. 11–23).

The robustness of spin coherence in **1** prompted us to investigate the spin distribution by CW and pulsed EPR methods.

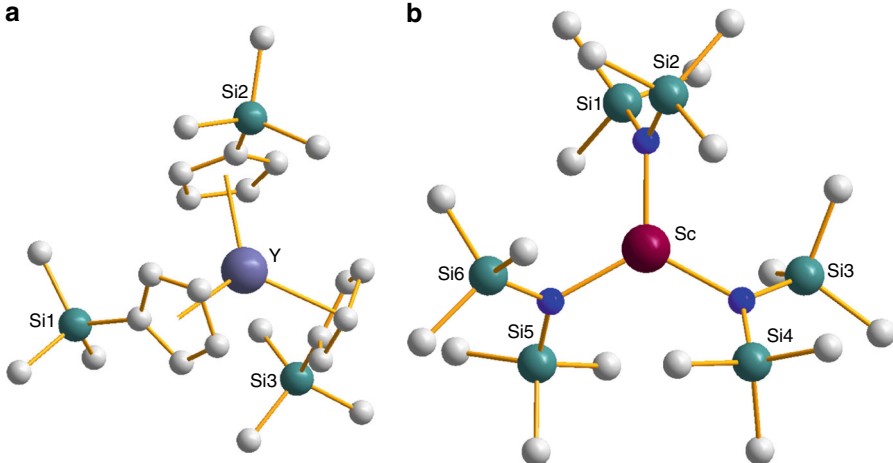

**Fig. 1** Molecular structures in the crystal. **a** The [Y(Cp′)$_3$]$^-$ anion in **1**. **b** The [Sc{(N(SiMe$_3$)$_2$)$_3$}]$^-$ anion in **4**. Grey unlabelled atoms are carbon; blue unlabelled atoms are nitrogen. H-atoms omitted for clarity

**Table 1 Observed *g*-values and hyperfine coupling constants for 1–4 in frozen solution at X-band (*ca*. 9.5 GHz)**

| Compound | *I* | Abundance (%) | *g* | \|*A*\| (MHz) |
|---|---|---|---|---|
| 1 | 1/2 | 99.9 | $g_z = 2.001$, $g_{x,y} = 1.986$ | $A_z = 98.6$, $A_{x,y} = 100.8$ |
| 2 | 7/2 | 97.0 | $g_z = 1.975$, $g_{x,y} = 1.945$ | $A_z = 1070$, $A_{x,y} = 1121$ |
| 3 | 7/2 | 99.9 | $g_z = 1.978$, $g_{x,y} = 1.935$ | $A_z = 400$, $A_{x,y} = 420$ |
| 4 | 7/2 | 100.0 | $g_z = 1.997$, $g_{x,y} = 1.964$ | $A_z = 640$, $A_{x,y} = 620$ |

We have quantified the electron spin density on the ligands with hyperfine sub-level correlation and electron nuclear double resonance (HYSCORE and ENDOR respectively;[19] see Fig. 3) measurements on **1** in THF solution. Weak hyperfine couplings to $^1$H and $^{13}$C nuclei are observed (Fig. 3, Supplementary Figs. 25–29; Table 2; the orientation of the Cp′ ring with respect to the molecular C$_3$ axis is shown in Supplementary Fig. 24), and are on the order of 1–2 MHz for the Cp′ ligands. We can be confident that we are not missing any large ligand hyperfine interactions (i.e. >4 MHz) because there is no resolution of $^1$H hyperfine in the CW EPR (which has narrow linewidths).

Pulsed EPR studies of **2–4** confirm this is a general phenomenon; as there was not an easily available diamagnetic host for these complexes we have had to restrict our measurements to the melting point of the THF solvent (90 K). In this temperature range the behaviour for $T_1$ for **1–4** is very similar. $T_m$ for **1–3** are almost identical in this range; compound **4** shows a fall in $T_m$ at the highest temperature measured compared with the other compounds.

**Electronic structure calculations**. Density functional theory (DFT) calculations[21] on the crystal structure of the anion in **1** give excellent agreement with experiment and are independent of the functional and basis set chosen (Table 2 and Supplementary Tables 10–13). The calculated $^{89}$Y hyperfine is dominated by the Fermi-contact term (i.e. electron density at the $^{89}$Y nucleus) and the spin-dipolar contribution is almost insignificant (hence the vanishing anisotropy) (Supplementary Table 14). The Fermi-contact term is dominated by the singly-occupied molecular orbital (SOMO) in which the unpaired electron resides (Fig. 4a), with small contributions from polarisation of core s-states. Lödwin analysis suggests 62% of the total spin density is metal-based, with the other 38% delocalised across the ligand atoms: the metal contribution breaks down as 7.5% s-based, 1.8% p-based and 52%

d-based. Despite the significant d-orbital contribution to the spin density, the hyperfine interactions are much more isotropic than predicted based on atomic wavefunctions[22], which would suggest $|A_z{}^Y - A_{x,y}{}^Y| \sim 29$ MHz, compared to the ca. 5 MHz calculated and 2 MHz observed here. The same conclusions can be drawn for an optimised structure of the anion in **1** or when the electrostatic potential of the crystalline environment is included (see Supplementary Fig. 30), indicating this is not a quirk of the crystalline geometry nor gas-phase calculations.

In order to understand the minimal g-anisotropy of **1**, we have conducted complete active space self-consistent field (CASSCF) calculations[23] using the RAS-probing technique[24] (see Supplementary Methods, Supplementary Figs. 31–33, Supplementary Tables 15–26) on the crystal structure of the anion in **1**. After including spin-orbit coupling, the g-values of the ground $S = 1/2$ state are $g_{x,y} = 1.989(1)$ and $g_z = 2.002$, in excellent agreement with the experimental data and DFT results (Table 1). The ground state SOMO is only 58% derived from Y-based functions suggesting that it is substantially delocalised (Supplementary Table 17). This Y contribution is dominated by the 4d$_z{}^2$ function (Fig. 4b and Supplementary Fig. 34), in good agreement with the spin density determined from DFT (Fig. 4a). Despite this however, the extensive delocalisation onto the ligands and significant Y s-orbital contribution dominate the spin Hamiltonian, giving an essentially isotropic hyperfine interaction. The result is that the anisotropic component accounts for only $(A_z - A_x)/A_{iso} \approx 2\%$ of the total hyperfine coupling.

Our CASSCF calculations show that the first excited state is only *ca.* 9000 cm$^{-1}$ above the ground state where the unpaired electron occupies a diffuse ligand orbital (largest Y contribution is 3% 4d, with 89% deriving from ligand functions; Supplementary Fig. 35 and Supplementary Table 18); calculations where the electrostatic potential of the crystalline environment is included show similar results, indicating that this is not the result of an

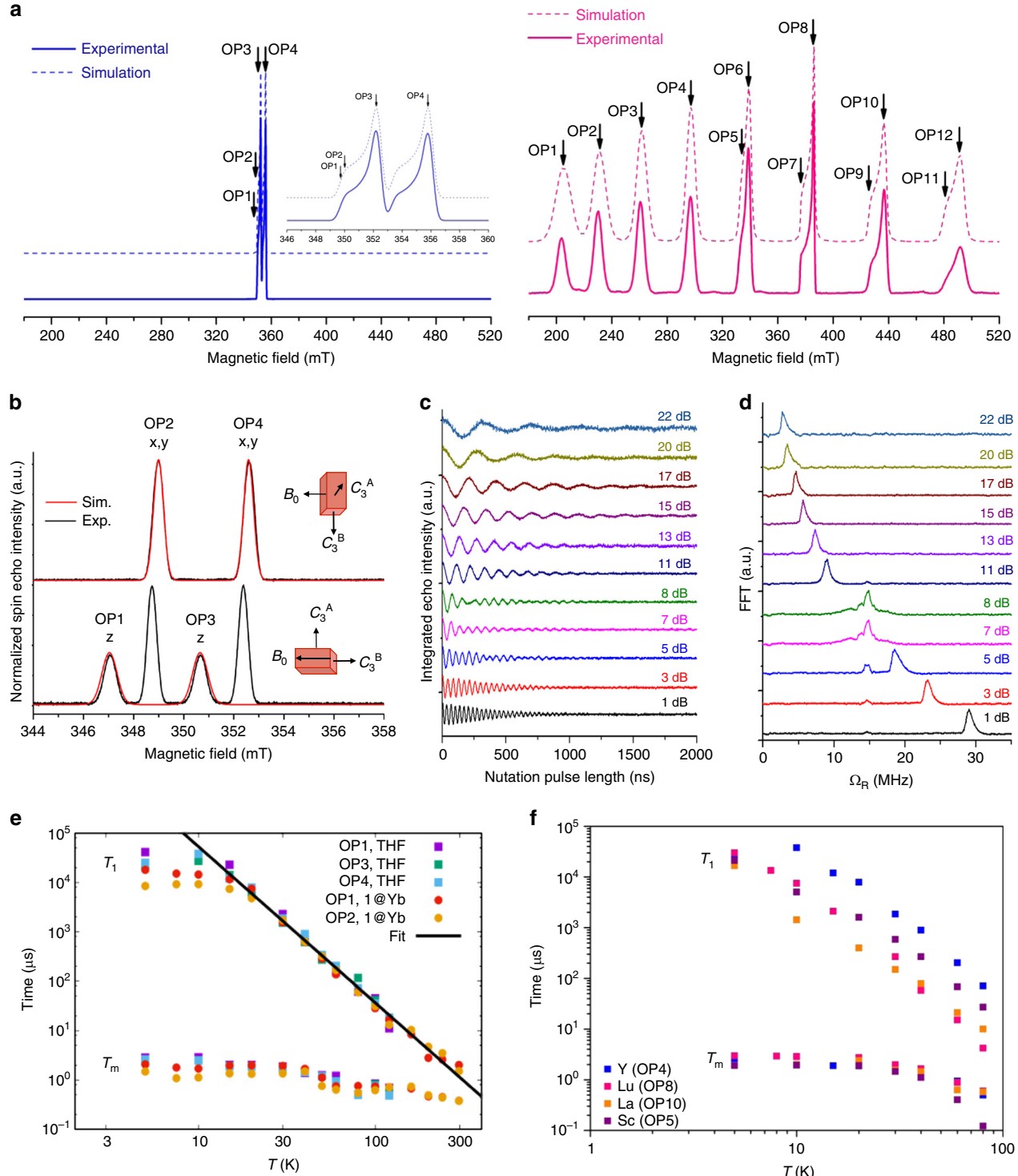

**Fig. 2** Echo-detected field-swept EPR spectra. Experimental (full) and simulated (dashed) EDFS spectra at 50 K and X-band (9.85 GHz) for a 10 mM THF solution of compounds **1** (blue) and **2** (pink), with the parameters in Table 1. **b** Experimental (black) and calculated (red) EDFS spectra at 200 K and X-band (9.75 GHz) for a single crystal of **1@5** at two selected orientations (Supplementary Figs. 3 and 4). Simulations are only for molecule **B**, such that only the resonances along the z-axis for molecule **B** are simulated in the lower figure (see Supplementary Methods for definition of molecule **B**). **c**, Rabi oscillations for **1@5** at 298 K and $B_0 = 347$ mT (OP1), acquired with different microwave attenuations (Supplementary Tables 5–9); **d** Fourier transforms of nutation data in **c**, giving the Rabi frequency (the 15 MHz signal is due to $^1$H ESEEM). **e** Relaxation times measured by pulsed EPR methods for **1** in two phases (frozen THF solution and single crystal **1@5**); black line is a simultaneous fit of all the $T_1$ data above 20K with $T_1 = C^{-1}T^{-n}$ where $C = 1.3(3) \times 10^{-8}$ $\mu s^{-1}$ $K^{-n}$ and $n = 3.16(5)$ (Supplementary Tables 1–4). **f** $T_1$ and $T_m$ dependence to 100 K for a single representative orientation of **1**, **2**, **3** and **4**. Observer positions OP1–OP12 mark the magnetic fields where time-dependent pulsed EPR experiments were performed

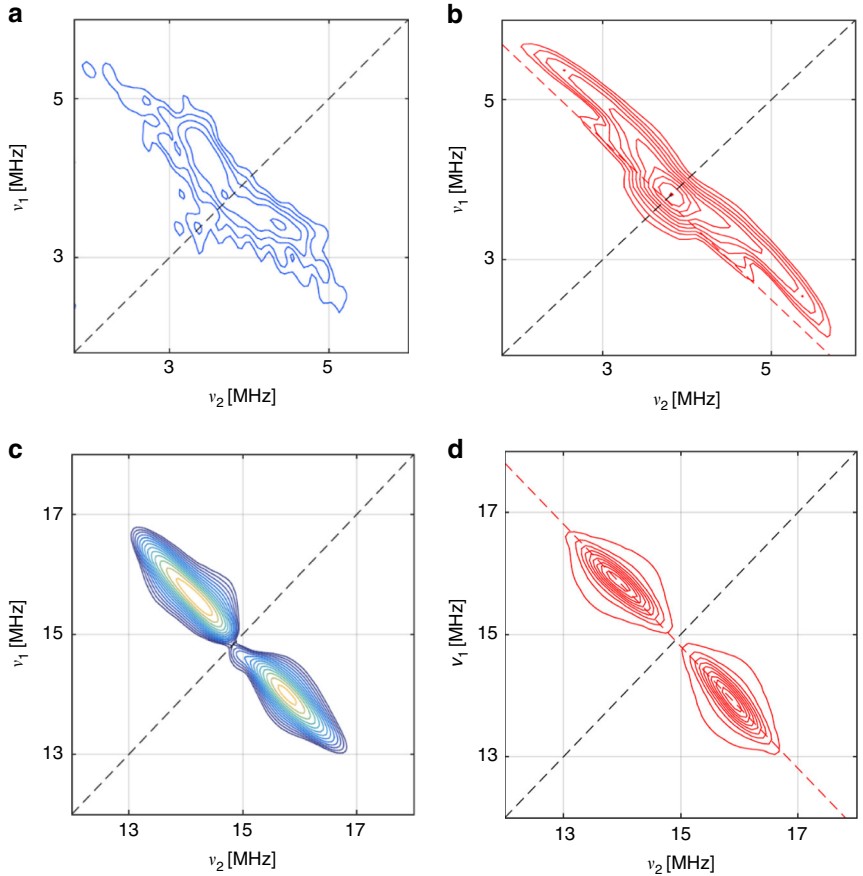

**Fig. 3** Pulsed EPR studies of a frozen solution of **1** (10 mM, THF). **a** $^{13}$C HYSCORE spectrum at 50 K and $B_0 = 352$ mT (OP3). **b** Simulation of the data in (**a**). **c** $^1$H HYSCORE spectrum at 50K and OP3. **d** Simulation of the data in (**c**). Simulations used EasySpin[32]

| Table 2 Experimental and calculated spin Hamiltonian parameters for 1 | | | |
|---|---|---|---|
| **Compound** | **T (K)** | **g** | **|A| (MHz)** |
| **1** (10 mM, THF) | 295 | $g_{iso} = 1.991$ | $|A_{iso}{}^Y| = 102$ |
| **1** (10 mM, THF) | 50 | $g_z = 2.001$, $g_{x,y} = 1.986$ | $A_z{}^Y = 98.6$, $A_{x,y}{}^Y = 100.8$ |
| | | | $A_{\parallel}{}^{C2/5} = 2.8$,[a] $A_{\perp}{}^{C2/5} = 0.4$ |
| | | | $A_{\parallel}{}^{C3/4} = 0.83$,[a] $A_{\perp}{}^{C3/4} = 0.3$ |
| ~2% **1@5** | 200 | $g_z = 1.999$, $g_{x,y} = 1.986$ | $|A_z{}^Y| = 99.5$, $|A_{x,y}{}^Y| = 101.3$ |
| DFT (gas-phase) | – | $g_z = 2.003$, $g_{x,y} = 1.990(1)$ | $A_z{}^Y = -93.86$, $A_{x,y}{}^Y = -88.7[4]$ |
| | | | $|A_{iso}{}^{C1-5}| = 1.1[8]$ |
| | | | $|A_{iso}{}^{H2-5}| = 2[1]$ |
| CASSCF (gas-phase) | – | $g_z = 2.002$, $g_{x,y} = 1.989(1)$ | – |

[a]For the parallel and perpendicular $^{13}$C hyperfines, the labels refer to the local principal axes, with the unique axis defined by the Cp' $\pi$-system, i.e., orthogonal to the molecular $z$ axis

unbound electron in the gas-phase (Supplementary Figs. 44–55 and Supplementary Tables 16 and 27–37). There are seven excited states less than 20,000 cm$^{-1}$ above the ground state, all having the unpaired electron in very diffuse ligand-based orbitals (Supplementary Figs. 35 to 41 and Supplementary Tables 18–24). This indicates that the electron is easily delocalised across the molecule, and explains why the ground state SOMO cannot be simply described as either a 4d$_z{}^2$ or 5s function on Y. Indeed, it is not until the ninth and tenth excited states (at ca. 34,000 cm$^{-1}$) that we find other metal-dominated MOs, which correspond mainly to the Y-based 4d$_{xz}$ and 4d$_{yz}$ functions (Supplementary Figs. 42 and 43, Supplementary Tables 25 and 26). The large energy gap to the 4d excited orbitals quenches the orbital angular momentum and this leads to the small $g$-anisotropy of the ground state.

## Discussion

The experimental data and computational results demonstrate without question that complex **1** has a very isotropic $S = 1/2$ ground state with weak spin orbit coupling (small $g$-shift and $g$-anisotropy) and s-orbital-dominated hyperfine interaction. The isotropic nature of the spin Hamiltonian is consistent with the observation that there is no significant variation in relaxation dynamics for different orientations, despite the 4d$_z{}^2$ admixture to the SOMO. We believe this is key to the unexpected relaxation properties of **1**.

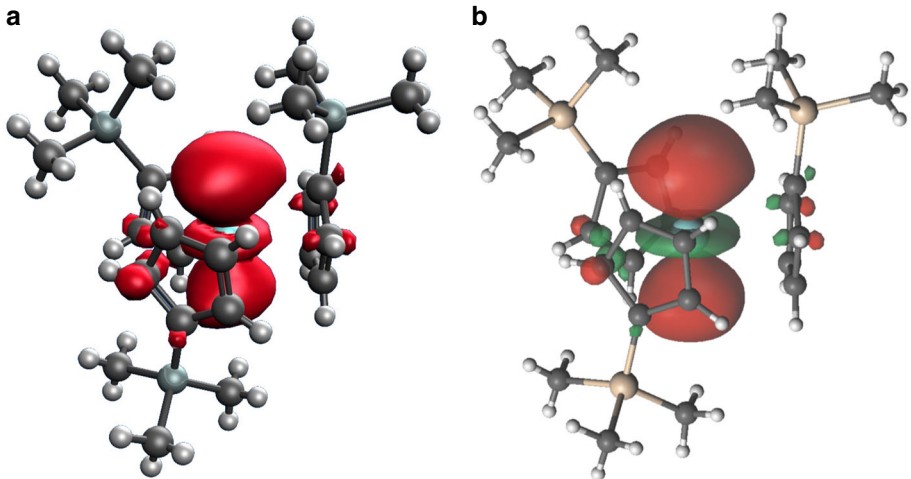

**Fig. 4** The calculated electronic structure of the anion in **1**. This shows: **a** spin density from DFT; **b** the SOMO (natural orbital for the ground $S = 1/2$ state) from CASSCF

The room temperature $T_1$ and $T_m$ of 2 and 0.4 µs, respectively, for **1** can be compared to the few other transition metal complexes that exhibit room temperature coherence (Fig. 5): [VO(Pc)] (Pc = phthalocyanine; 0.001% dilution) has $T_1$ and $T_m$ of 1 and 0.8 µs, respectively;[25,26] [VO(dmit)$_2$] (dmit = 1,3-dithiole-2-thione-4,5-dithiolate; 5% dilution) has $T_1$ and $T_m$ of 3 and 1 µs, respectively;[11] [Cu(mnt)$_2$] (mnt$^{2-}$ = maleonitriledithiolate; 0.001% dilution) has $T_1$ and $T_m \approx 0.6$ µs[8]. The latter two examples exploit nuclear-spin-free ligands, while the first example has a highly rigid phthalocyanine ligand framework. Neither of these features is present in **1**, nor have we optimised the magnetic dilution.

A consequence of the small anisotropy and SOC in **1** is a longer than expected $T_1$, brought about by reducing the effect of molecular vibrations in this non-rigid molecule. This becomes more important at higher temperatures where $T_m$ does not become limited by $T_1$ up to 300 K. The delocalised nature of the SOMO may also contribute to lengthening $T_1$. In the few literature examples where coherence is observed at room temperature, both [VO(Pc)] and [Cu(mnt)$_2$] reach the $T_m = T_1$ limit (Supplementary Fig. 10).

This still does not explain the surprisingly long $T_m$ values given the nuclear spin-rich, methyl group rich and flexible coordination geometry of **1**. It is possible that the ligand nuclei are close enough to the paramagnetic metal ion such that their participation in nuclear spin diffusion is limited: the diffusion-barrier radius has been estimated at ca. 6 Å in studies of nitroxyl and trityl radicals:[18] the Y-H(Cp) distances are 3.1–3.3 Å in **1**, and the Y-H(Me) distances lie in the range 3.8–6.4 Å. In addition, even when nuclear spin diffusion dominates decoherence, it can be enhanced by small amplitude librations that result in significant changes in the resonance frequency[18]. This effect is minimised near principal axes where the resonance frequency is least sensitive to orientation: the resonance condition for the essentially isotropic **1** is necessarily flat with respect to molecular orientations, hence this mechanism will be largely quenched. This will be true of any $T_m$ mechanism that works by dynamic averaging of $g$ and $A$ anisotropy.

We conclude that the robust relaxation behaviour of **1** is intrinsically associated with the isotropic electronic structure, which is due to quenching of the orbital angular momentum in the ground state. The near isotropic magnetic properties mean that **1** is behaving spectroscopically like an atomic $^2$S-state. Preliminary solution phase studies (Fig. 2f) and calculations (Supplementary Table 14) of **2–4** show this is a general phenomenon; **4** contains amide ligands, not Cp ligands, and hence the phenomenon is due to engineering the electronic structure of the metal and not due to a specific ligand.

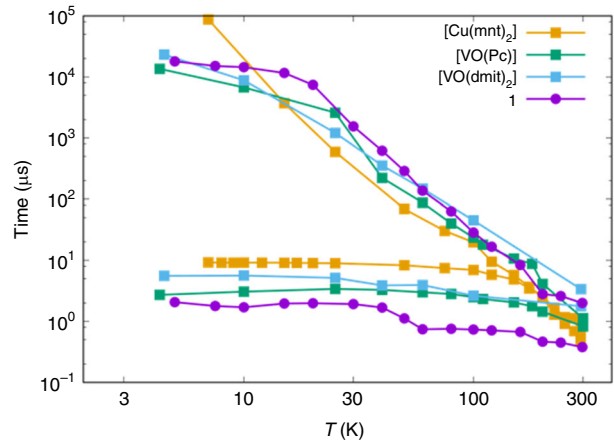

**Fig. 5** Spin-lattice and phase memory times for molecular spin qubits. Comparison of $T_1$ (upper) and $T_m$ (lower) as a function of temperature for [Cu(mnt)$_2$] (0.001% solid-state dilution), [VO(Pc)] (0.001% solid-state dilution), [VO(dmit)$_2$] (5% solid-state dilution), and ~2% **1@5**

There are variations between the behaviour of **1–4**; the $g$-values are most isotropic and nearest the free electron $g$-value for **1** (Table 1). This also correlates with $T_1$ being longest for **1** at any temperature measured. Compounds **2** and **3** are very similar to each other, and it could be assumed that in both cases the spin-orbit coupling would be larger than in **1** due to the higher atomic number (Lu and La vs. Y); we suspect this is the origin of the larger $g$-shift and shorter $T_1$ in both compounds. Compound **4** falls outside this trend, as Sc is lighter than Y and yet has a larger $g$-shift and shorter $T_1$ than **1**, and this could be connected with some influence of the different ligand. A future synthetic target is a *tris*-cyclopentadienyl scandium(II) compound to make the direct comparison.

The utility of a $^2$S$_{1/2}$-like electronic ground state has long been exploited in trapped atom/ion qubits studied at milli-Kelvin temperatures[27–29]. Studies on alkali metal atoms[30] and silver atoms[31] by pulsed or CW EPR methods give similar $T_1$ values to those found here in a molecular anion, however such studies are limited to the melting point of the matrices used, which is always well below room temperature. It is a challenge to produce a pure $^2$S state within a molecule and the next target to increase coherence times must be to engineer both the electronic structure of the metal ion and the vibrational modes of the ligand.

## Methods

**Continuous-wave EPR measurements.** Continuous-wave (CW) electron para-magnetic resonance (EPR) spectra of solution samples of **1** (Fig. 2a) were recorded on either a Bruker EMX 300 or a Bruker ElexSys E580 EPR spectrometer operating at X-band (ca. 9.4–9.8 GHz) and variable temperatures. CW EPR spectra of oriented single crystals of ~**2% 1@5** (Fig. S3) were collected with a Bruker ElexSys E580 instrument operating at ca. 9.7 GHz and varied temperatures and equipped with a goniometer that allowed controlled crystal rotation. An identical setup was used to measure the same crystal by echo-detected pulsed-EPR methods. Crystals were indexed by X-ray crystallography to determine the precise orientation of the crystallographic axes.

**Pulsed EPR measurements.** Pulsed EPR spectra were recorded with a Bruker ElexSys E580 instrument equipped with either a MD5 or a MD4 resonator and operating at ca. 9.7 GHz and various temperatures. Solution samples of different concentrations (2, 5 and 10 mM in THF) were investigated to check reproducibility and to achieve an acceptable signal-to-noise response in HYSCORE and ENDOR experiments.

**DFT and CASSCF calculations.** All DFT calculations were performed using ORCA 4.0.0.2[21] with the unrestricted Kohn-Sham formalism on the $S = 1/2$ ground state of the anions in **1–4**. For geometry optimisations we started with the crystal structures. State-averaged CASSCF calculations for the anion in **1** were performed with MOLCAS 8.0[23].

## Data availability

Supplementary information is available in the online version of the paper. Reprints and permissions information are available online at www.nature.com/reprints.

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

## Acknowledgements

This work was supported by the European Union, via a doctoral scholarship to A.-M.A. (FP7-PEOPLE-2013-ITN MAGIC Initial Training Network; grant agreement no. 606831), and MOLSPIN COST Action CA15128; the U.S. National Science Foundation (CHE-1565776); the Ramsay Memorial Trust (fellowship to N.F.C.), the Leverhulme Trust (fellowship to F.T. reference RF-2018-545/4); the Engineering and Physical Sciences Research Council (DTG doctoral scholarships to L.N. and A.K.K.), The University of Manchester, The University of California, Irvine, and the UK National EPR Facility and Service. We thank Dr Joe McDouall for insightful conversations, and Profs. Joris van Slageren and Roberta Sessoli for providing experimental data for the [Cu(mnt)₂], [VO (Pc)] and [VO(dmit)₂] complexes.

## Author contributions

A.-M.A. and L.N. and F.T. collected and interpreted the EPR spectroscopy data, with support from A.K.K. and E.J.L.M. D.H.W. and D. N. H. synthesised and characterised the compounds under the supervision of W.J.E. C.A.P.G. made the EPR samples. N.F.C. performed and interpreted the DFT and CASSCF calculations. W.J.E. and R.E.P.W. proposed the initial concept. F.T. designed the work and supervised A.-M.A. R.E.P.W., F.T., A.-M.A., N.F.C. and E.J.L.M. wrote the manuscript, with contributions from all co-authors.
