## [Transparent Peer Review File · Nature Communications]

Editorial Note: This manuscript has been previously reviewed at another journal that is not operating a transparent peer review scheme. This document only contains reviewer comments and rebuttal letters for versions considered at Nature Communications. Mentions of the other journal have been redacted.

REVIEWERS' COMMENTS:

Reviewer #1 (Remarks to the Author):

The authors have removed the incorrect/misleading statements contained in the previous version submitted to [redacted]. The revised discussion of the experiments is now much more grounded. Indeed, it is clear that the preparation of the revised manuscript stimulated some new lines of thinking. Certainly, the discussion towards the end of the manuscript has been extensively revised.

A nagging concern that I still have is that the overall findings remain poorly understood/explained. It was never my intention to lead the authors into such a trap. Nevertheless, the comparison in Figure S10 of the SI shows that the present compound 1 is not really superior to the other compounds in terms of spin-lattice (T_1) relaxation. Yet this claim is made very strongly in the main text (lines 227 to 230). The authors instead chose to plot T_m/T_1 in the main article (new Figure 5), where their compound 1 clearly stands out relative to the others (also very apparent in Fig. S10). However, the contrast appears to be due mainly to differences in T_m , as opposed to T_1 . This does get back to comments made in previous rounds of reviewing, and remains the biggest mystery of this study. The authors at least make some attempt to speculate as to possible reasons why T_m is surprisingly long in the paragraph starting on line 216. This section includes a brief discussion of the effects of small amplitude librations, but no reference is given.

Overall, this is a very intriguing study. As I have maintained from the outset, the experiments and analysis are of the highest standard. I am fairly certain that the paper will attract considerable attention, but perhaps not for the originally anticipated reasons. However, I am torn as to whether it meets the standards for publication in a Nature journal.

Since the authors seem determined to publish this work [redacted] I would ask the following.

Take one last step back and look at the big picture. The paper was clearly written originally with some different lines of thinking in mind. This resulted in the introductory paragraph (abstract?) centered around reductive/protective approaches to the decoherence problem in molecular spin qubits. I am not sure that this now fits given that it is not at all clear why T_m is so long in compound 1. I believe that the authors made a genuine attempt to improve discussion of the experiments in the revised version of the manuscript. I feel that the abstract could be further edited to reflect the new discussion. I would also ask that they modify the discussion of their T_1 with respect to other examples in the molecular spin qubit literature. In the end, this interesting work needs to see the light of day.

REVIEWERS' COMMENTS:

Reviewer #1 (Remarks to the Author):

The authors have removed the incorrect/misleading statements contained in the previous version submitted to [redacted]. The revised discussion of the experiments is now much more grounded. Indeed, it is clear that the preparation of the revised manuscript stimulated some new lines of thinking. Certainly, the discussion towards the end of the manuscript has been extensively revised.

A nagging concern that I still have is that the overall findings remain poorly understood/explained. It was never my intention to lead the authors into such a trap. Nevertheless, the comparison in Figure S10 of the SI shows that the present compound 1 is not really superior to the other compounds in terms of spin-lattice (T_1) relaxation. Yet this claim is made very strongly in the main text (lines 227 to 230). The authors instead chose to plot T_m/T_1 in the main article (new Figure 5), where their compound 1 clearly stands out relative to the others (also very apparent in Fig. S10). However, the contrast appears to be due mainly to differences in T_m , as opposed to T_1 . This does get back to comments made in previous rounds of reviewing, and remains the biggest mystery of this study. The authors at least make some attempt to speculate as to possible reasons why T_m is surprisingly long in the paragraph starting on line 216. This section includes a brief discussion of the effects of small amplitude librations, but no reference is given.

Overall, this is a very intriguing study. As I have maintained from the outset, the experiments and analysis are of the highest standard. I am fairly certain that the paper will attract considerable attention, but perhaps not for the originally anticipated reasons. However, I am torn as to whether it meets the standards for publication in a Nature journal.

Since the authors seem determined to publish this work [redacted] I would ask the following.

Take one last step back and look at the big picture. The paper was clearly written originally with some different lines of thinking in mind. This resulted in the introductory paragraph (abstract?) centered around reductive/protective approaches to the decoherence problem in molecular spin qubits. I am not sure that this now fits given that it is not at all clear why T_m is so long in compound 1. I believe that the authors made a genuine attempt to improve discussion of the experiments in the revised version of the manuscript. I feel that the abstract could be further edited to reflect the new discussion. I would also ask that they modify the discussion of their T_1 with respect to other examples in the molecular spin qubit literature. In the end, this interesting work needs to see the light of day.

Answer:

We again thank the reviewer for this excellent job. We have rewritten the abstract completely and moved some of this material into the introduction, which has also been rewritten. The new abstract is 148 words long. We think the paper is significantly improved by this change, so thank you.

We have swapped Figures S10 and Figure 5 as we realised, on reading the review, that the new Figure 5 contains more easily assimilated information than the previous Figure 5.

We have also rewritten the section immediately around Figure 5 to soften the claims concerning T_1 .

During revision we became concerned at one aspect of the electronic structure calculation and we revisited this. The outcome of the newly revised calculation is very close to the calculation we reported in our original submission to [redacted]; these calculations were not questioned by reviewers. This has led to minor changes to numbers in the paragraph that begins at the bottom of P8. The revised calculation does not change any of the conclusions of the paper.

We have also added in Introduction, Results, sub-headings (Electron Paramagnetic Resonance Spectroscopy and Electronic Structure Calculations) and Discussion.